# Remote Neuropsychological Intervention for Developmental Dyslexia with the Tachidino Platform: No Reduction in Effectiveness for Older Nor for More Severely Impaired Children

**DOI:** 10.3390/children9010071

**Published:** 2022-01-05

**Authors:** Maria Luisa Lorusso, Francesca Borasio, Massimo Molteni

**Affiliations:** 1Unit of Child Psychopathology, Scientific Institute IRCSS E. Medea, 23842 Bosisio Parini, Italy; francesca.borasio@lanostrafamiglia.it (F.B.); massimo.molteni@lanostrafamiglia.it (M.M.); 2Department of Psychology, Catholic University of the Sacred Heart, 20123 Milano, Italy

**Keywords:** dyslexia, remote intervention, children, age, severity, improvement, follow-up

## Abstract

Tachidino is a web-platform for remote treatment of reading and writing disorders. A total of 91 children with developmental dyslexia and/or dysorthographia participated in the present study and received Tachidino treatment. The purpose of the study was to compare results obtained after four weeks treatment and a six-month follow-up in older versus younger children and in more versus less severely impaired children (separately subdividing them according to reading speed, reading accuracy, and writing accuracy). The results showed no difference in improvement for reading accuracy and speed in the three age groups, but children below 9 years improved more than older children in writing accuracy. Regarding severity groups, children with more severe initial impairments improved more than children with less severe impairments. Additionally, the results were confirmed after controlling for spurious effects due to use of Z-scores and regression to the mean. The findings are discussed in terms of their theoretical and practical implications.

## 1. Introduction

Developmental dyslexia is a specific learning disorder affecting reading and spelling that is not due to low intelligence, sensory or neurological damage, or poor educational opportunities [1,2]. Recent research has suggested a multiple-deficit model of dyslexia that could provide a better description of the profiles and risk factors associated with learning difficulties [3,4]. This kind of comprehensive, multi-factor model should be considered to provide an individually tailored treatment program, focused on children’s multiple needs [5,6].

Tachidino is a fully automated training programme for reading and writing disorders [7]. It is a web-based platform including a training software, along with systems managing clients’ and professionals’ data and interfaces. The training software incorporates multi-componential principles, specifically Visual-Attentional Training [8,9] and Visual Hemisphere-Specific Stimulation (VHSS) according to Bakker’s Balance Model [10,11] revisited for adaptation to current Italian clinical practice [12]. VHSS stimulates selectively one visual hemisphere to improve reading: this treatment, in isolation, has already been demonstrated to improve both reading fluency and accuracy in dyslexic children [11,12,13,14,15,16,17]. The Visual-Attention training is inspired by Action Video Games (AVG), which are characterized by an emphasis on peripheral processing and global perception of stimuli moving at high speed and that are spatio-temporally unpredictable [18,19]. Some studies on children with dyslexia showed that AVGs improve visual-attention abilities, spatial cognition, auditory spatial attention, response time speed, word recognition and phonological decoding efficiency [8,18,20,21].

Tachidino combines the two types of training: in the present training programme, the child is first required to identify and select a moving object among other similar moving objects (visual attention training component) and then to decode or encode words or short text sequences (hemisphere-specific decoding strategies). The text may consist of visually, tachistoscopically presented words/nonwords/combinations or by the same verbal stimuli, auditorily presented through text-to-speech-software. All parameters, such as the laterality of the stimulus, the type of exercise and the list of stimuli, are adapted to the child’s characteristics [7]. Auditory presentation was especially added to the program in order to stimulate writing abilities, requiring phoneme-to-grapheme conversion. Indeed, it has been shown [12] that rapid, central rather than lateralized stimulation was the most effective type of stimulation for the improvement of writing abilities, suggesting that writing processes (different from reading) profit more from bilateral stimulation and thus possibly from inter-hemispheric integration. 

The remote use of Tachidino allows the maximization of effectiveness in improving reading skills by optimizing the duration and flexibility of the intervention, making the program available to a large group of patients without increasing the costs to health services. Remote treatments for developmental dyslexia can improve reading speed and accuracy after only a few weeks and may foster automatization of the reading process [22]. Moreover, intervention delivered via telepractice appears feasible and engaging, and its effectiveness seems to be comparable to face-to-face programs, although stronger evidence is still needed [23].

There is a general agreement that early identification and intervention are crucial to ensure that children with developmental dyslexia can maximise their educational potential and ameliorate deficits in reading skills [24,25]. However, most studies focused on reading problems in younger children and less is known about training for older children [6]. An Italian study [26] explored whether there may be differences in gains on reading accuracy and fluency between younger and older children with dyslexia (attending primary and secondary school: young group—from the end of the second grade to the end of the fourth grade, versus old group—from the sixth to the eighth grade). These authors found no differences in improvements of reading fluency and accuracy between the two age groups. Moreover, they observed equivalent changes in younger and older children, even if at pretreatment younger children were less accurate and fluent with respect to older children. The demonstration that reading skills may be improved at relatively older ages is an encouraging perspective for children receiving a late diagnosis or treatment. Further studies showed that older children could obtain gains comparable to those of younger students: there is no evidence that older children cannot benefit from specific training [27]. The authors of a recent study evaluated the impact of causes, correlates, and consequences of dyslexia in predicting the outcome of intervention [28]. Among potential impact factors was the severity of reading problems. It was found that children with higher pretreatment word reading skills demonstrated greater improvements compared with peers with more severely impaired reading abilities. These results were in line with those of previous studies e.g., [29].

Based on these premises, it is commonly believed that younger and/or less impaired children are likely to benefit more from intervention on their reading or writing/spelling difficulties. Nonetheless, the results of some studies showing large, clinically relevant improvement in older children, adults, and more severely impaired individuals [26,30,31,32] in addition to clinical experience led us to reconsider such beliefs.

The purpose of the present study was thus to compare the gains in reading speed and accuracy, and in writing accuracy of different groups of children with a diagnosis of developmental dyslexia. To assess possible age-related differences on the outcomes of the treatment, children were subdivided into three different age groups of similar size, representing different stages of reading acquisition: initial, intermediate, and consolidated. Moreover, to assess the impact of the severity of the impairments on the gains, children were divided into three groups characterized by different levels of impairment in reading and writing. The gains obtained with the treatment for dyslexia were compared both after one month of Tachidino treatment and at follow-up six months after discontinuation of the treatment. Based on clinical experience and on the (few) studies described above, we expected that all groups, irrespective of age and impairment severity, would significantly benefit from the treatment and that no differences would emerge between groups.

## 2. Materials and Methods

### 2.1. Participants

A total of 91 children (54 male) aged between 7 and 14 years (mean age = 9.44 years, SD = 1.41) were recruited for this study. Participants had to fulfil the following inclusion criteria: (a) having been diagnosed with Specific Reading Disorder (ICD-10 codes: F81.0 or F81.3) on the basis of standard inclusion and exclusion criteria (ICD-10, WHO, 2011) (at least one Z-score concerning reading/writing speed and/or accuracy below −2); (b) absence of comorbidity with other psychopathological conditions (whereas comorbidity with other learning disorders and/or ADHD was allowed); (c) not having been involved in other clinical intervention programs for learning disorders in the past year. Since the children had been diagnosed, on average, two to three months before the start of the treatment, persistence of reading/writing impairments at pre-test assessment was considered as a confirmation of the stability of the disorder prior to intervention.

Children were selected among patients referred to the IRCCS “E. Medea” and the recruitment took place between January 2018 and January 2020. The present study analyzes data from an observational study for the systematic description of treatment results in a whole, unselected cohort of children with reading and writing disorders in charge at the Institute. The study was approved by the Local Ethics Committee according to the declaration of Helsinki. All parents were informed of the study goals and procedures and their written informed consent was obtained before the beginning of the study. Eighty children (46 male), aged between 7 and 14 (mean age = 9.48 years, SD = 1.44), participated in Time 1, Time 2, and Time 3 assessments. Eleven (12%) participants were no longer available at the time of follow-up, Time 3, due to family reasons not related to the treatment (moving, previous contacts no longer active etc.); performance at pre-test and improvement expressed as D-scores were compared in the two groups, children present and absent at follow-up, and showed no significant differences, all *p* values > 0.15. A sensitivity analysis (G-power) showed that a sample of 87 participants allows detection of an effect size of 0.17 in a three group, one-tailed, repeated measures ANOVA, within-between interaction, with a power of 0.80. Based on previous results obtained with VHSS [12,13], this effect size was considered acceptable.

The data collection at Time 1, Time 2, and Time 3 of the Tachidino treatment is registered in ClinicalTrials.gov (Code NCT04382482) as an observational study.

### 2.2. Neuropsychological Tests

The tests commonly employed in the assessment of reading disorders in Italy were used. The results of the tests are expressed as Z-scores according to age norms. The following tests were administered to children in the pre- and post-test and follow-up sessions.

Text reading: “Prove di rapidità e correttezza nella lettura del gruppo MT” (“Test of speed and accuracy in reading, developed by the MT group”) [33]. This test assesses reading abilities for meaningful texts. It provides separate scores for speed and accuracy. Texts increase in complexity with grade level, and norms are provided for each text. Validity and reliability for the MT text reading test are reported to be satisfactory without further specifications [33]. For the last published version of the test, all test-retest reliability coefficients are above 0.59 and inter-rater reliability coefficients are 1.0 for speed and 0.99 for accuracy [34].Single word/non-word reading: “DDE-2: Batteria per la Valutazione della Dislessia e Disortografia Evolutiva-2” (Assessment battery for Developmental Reading and Spelling Disorders-2) [35]. The battery assesses speed and accuracy (number of errors) in reading word lists (4 lists of 24 words) and non-word lists (3 lists of 16 non-words), and provides grade norms from the second to the last grade of junior high school. The test has acceptable reliability (mean test-retest coefficients are 0.77 for speed and 0.56 for accuracy).Single word/non-word writing: two writing-to-dictation tasks were taken from the DDE-2 battery, giving accuracy scores according to age norms in writing 48 words and 24 non-words. Although these tests are commonly employed for diagnosis of spelling disorders, no specific reliability and validity data are available.

### 2.3. Procedure

A test-training-retest follow-up experimental design was carried out. All children were tested individually at three time points: Time 1, and Time 2 immediately before and after treatment, while Time 3 occurred 6 months after discontinuation of the treatment. The children’s reading and writing skills were assessed by a professional psychologist specialized in cognitive assessment; the psychologist in charge of the assessment was different from psychologists in charge of intervention. Children took part in the remote intervention program for an average total time of 14 h (range 12/18) over a maximum of four weeks. The training does not have a fixed working schedule, so as to adapt to the child’s rhythms and attention capacity, but the children are encouraged to work at least 4–5 days per week in sessions of 20–30 min, possibly repeated during the same day, and to read about 20–24 lists of words per week for three/four weeks (the average number of total lists was 73). The exact duration depends on the child’s reading level and is about 4–6 h per week. The program includes one pre-treatment meeting in order to define the dyslexia subtype, to demonstrate Tachidino use and programming the first activities, and one intermediate phone call to motivate and monitor correct use of the program.

### 2.4. Treatment

The Tachidino intervention program aims at improving reading through improvement of both decoding strategies and visuo-spatial attentional abilities. Indeed, in each trial of training the child is requested to perform both visual-spatial and word recognition tasks. Firstly, in order to receive the stimulus to be read or listened to, the child is required to recognize the target candy (a spiral candy) among various candies (distractors) and press the spacebar at the exact moment the target candy is crossing a circle target (fixation point). This procedure allows for the control of fixation for precise stimulation of the visual hemifields connected to a certain hemisphere, and also constitutes the visual-attentional training component. If the bar is pressed in the target timeframe and in correspondence of the target candy, the word (or nonword, or word combination) to be decoded/encoded is immediately presented (either visually, in the desired position, or auditorily), and the child is asked to either write the word on the keyboard or re-order a sequence containing all the correct graphemes in random order. This part constitutes the hemisphere-specific stimulation component.

All visual stimuli are presented at tachistoscopic speed to a visual hemifield in order to selectively stimulate the contralateral hemisphere, or they may also be flashed in the center of the computer screen, involving both hemispheres simultaneously. The visual hemisphere-specific stimulation is based on a revisit of Bakker’s ‘Balance model’ [10,11,12,13]. Each child was classified as a P-, L-, or M- type dyslexic reader based on the persistent over-reliance on specific reading strategies, on reading speed, and on the pattern of reading errors. More precisely, each child could be included in one of the three following subtypes:P-type (decoding strategies based on accurate perceptual analysis mainly supported by the right hemisphere-RH, resulting in slow but relatively accurate reading) if reading speed is at least 1 SD below age mean and the proportion of time-consuming errors over total errors is ≥60%;L-type (anticipation strategies based on linguistic abilities and mainly supported by the left hemisphere-LH, resulting in relatively fast but inaccurate reading) if reading speed is no more than 1 SD below age mean and the proportion of substantive errors over total errors is ≥60%;M-type (who strive to use both kinds of strategies but do so inefficiently, resulting in both slow and inaccurate reading) in all other cases (when both error types are present in similar proportion and/or when child is both slow and inaccurate in reading).

The tachistoscopic presentation of visual stimuli depended on classification, and selectively stimulated either the RH (additionally requesting RH-specific perceptual analysis using visually complex materials and/or error detection and correction tasks) or the LH (additionally requesting LH-specific linguistic anticipation using linguistically inter-related materials and/or anticipation/completion tasks). M-type dyslexic readers received the stimulation of the RH first and of the LH at a later stage of treatment, following the stages of natural reading acquisition according to the Balance Model. Central stimulation (and/or auditory presentation) was chosen when the target was to improve writing abilities.

Auditory stimuli were presented through Google text-to-speech synthesis, at the desired speed and pitch according to the child’s needs. During auditory presentation of the words, the child is encouraged to extract phonological information from the input, operate phoneme-to-grapheme conversion and match the auditory string with the written visual string (to be either written by the child or reconstructed based on given sequences of the correct graphemes randomly ordered). In the case of auditory stimulation, the hemisphere-specific part is represented in the choice of the material (e.g., low frequency, concrete, highly imageable words for RH stimulation vs. high-frequency, semantically interconnected, abstract words for the LH) and tasks (e.g., correcting errors in word spelling based on auditory input vs. writing or completing semantically related words), addressing either precise (de)coding strategies supported by the RH or anticipation strategies supported by the LH. Auditory presentation is to be considered as a secondary aspect in the training, but relevant for children whose main impairments are in spelling/writing more than in reading skills.

The therapist programs and monitors remotely the child’s activities, either in real time or at a different moment, and defines the graphic background, the laterality of the stimuli, the type of exercise (read/write, read/correct, listen/write, listen/correct), the lists of stimuli, exposure times, the characteristics of the font (type, size, spacing, color) and of the speech synthesis (speed and pitch) [7].

### 2.5. Data Analysis

Three global Z-scores were computed, for pre-, post-test, and follow-up:

(1) global reading speed score, the average of speed Z-scores in text, word and nonword reading, (2) global reading accuracy score, the average of accuracy Z-scores in text, word and nonword reading, and (3) global writing accuracy score, the average of accuracy Z-scores in word and nonword writing to dictation. Subsequently, difference scores between the post-test and the pre-test, the follow-up and the post-test, and the follow-up and the pre-test global scores were calculated, expressing training-related changes.

In order to compare the effectiveness of treatment in different age ranges, children were divided into three age groups: Group 1—younger than 9 years (*n* = 27), Group 2—between 9 and 10 years (*n* = 42), Group 3—11 years old and older (*n* = 22). Moreover, performances were compared between children with different impairment severity levels (degree of reading and writing impairment). In particular, children were divided into those with a performance in the target ability (in turn, reading speed, reading accuracy, and writing accuracy) of -1 Z-scores or higher (Group a), between -3 and -1 Z-scores (Group b) and of -3 Z-scores or lower (Group c). Depending on the specific individual profile, the same child could be included in different groups for different parameters (e.g., one child may have been included in the very severely impaired group with respect to reading speed, to the moderately impaired group for reading accuracy, and in the less severely impaired group for writing accuracy -n are reported in Table 1).

In order to assess the effectiveness of treatment and the possible treatment-by-group interactions, repeated measures ANOVAs, with age group or severity group as between-subjects factor, and treatment (pre-treatment/post-treatment) as within-subjects factor. Sporadic data are missing due to technical problems during data collection and recording.

## 3. Results

### 3.1. Treatment Related Changes: Comparison between Pre- and Post-Test Assessment

#### 3.1.1. Comparison between Age Groups

Comparing pre- and post-test performances with repeated measures ANOVAs with treatment as a within-subject factor and age group as a between-subject factor, a significant main effect of treatment on global reading speed, global reading accuracy, and global writing accuracy confirmed treatment effectiveness (F(1, 88) = 69.95, *p* < 0.001, η^2^_p_ = 0.443; F(1, 88) = 69.12, *p* < 0.001, η^2^_p_ = 0.440; F(1, 85) = 46.58, *p* < 0.001, η^2^_p_ = 0.354, respectively). At post hoc analyses, improvements were significant for all measures within each group (all *p* < 0.019).

For global reading speed and global reading accuracy the analyses showed no significant treatment x age interactions (all *p* > 0.27). However, a significant treatment x age interaction emerged (F(2, 85) = 9.43, *p* < 0.001, η^2^_p_ = 0.182) along with a significant age effect (F(2, 85) = 4.38, *p* = 0.015, η^2^_p_ = 0.094) on global writing accuracy. Tukey post-hoc analysis conducted on the age groups revealed a significant difference between Group 1 (younger than 9 years) and Group 2 (between 9 and 10 years) (*p* = 0.031), and Group 1 and Group 3 (11 years old and older) (*p* = 0.030), indicating that the treatment produced significantly greater improvements in the youngest group of children with respect to both other groups.

To avoid possible spurious effects due to the reduction in variance for reading scores at older ages, a repeated-measures ANOVA was conducted using raw scores instead of z-scores. In particular, syllables/second in text reading was used as a parameter expressing reading speed. Text reading speed was found to be significantly higher after treatment (F(1, 88) = 182.82, *p* < 0.001, η^2^_p_ = 0.675). A post-hoc analysis on syllables/second revealed a significant difference between Group 1 (younger than 9 years) and Group 2 (between 9 and 10 years) (*p* = 0.034). Moreover, a significant treatment x age interaction was found (F(2, 88) = 3.64, *p* = 0.030, η^2^_p_ = 0.182) together with a significant effect of age (F(2, 88) = 8.25, *p* = 0.001, η^2^_p_ = 0.158). At post-hoc analysis, a significant difference emerged between Group 1 (younger than 9 years) and Group 2 (between 9 and 10 years) (*p* = 0.008), and Group 1 and Group 3 (11 years old and older) (*p* = 0.001).

Table 1 presents descriptive results at pre- and post-test assessments.

#### 3.1.2. Comparison between Severity Groups

Pre- and post-test performances in global reading speed were compared across groups differing in reading speed impairment severity, global reading accuracy was compared across groups differing in reading accuracy impairment severity, and global writing accuracy was compared across groups differing in writing accuracy impairment severity. A repeated-measures ANOVAs showed significant main effects of treatment for global reading speed, global reading accuracy, and global writing accuracy (F(1, 88) = 97.2, *p* < 0.001, η^2^_p_ = 0.525; F(1, 88) = 92.81, *p* < 0.001, η^2^_p_ = 0.513; F(1, 85) = 90.99, *p* < 0.001, η^2^_p_ = 0.517, respectively). A significant treatment x severity effect was found for global reading speed (F(2, 88) = 10.31, *p* < 0.001, η^2^_p_ = 0.190), global reading accuracy (F(2, 88) = 15.24, *p* < 0.001, η^2^_p_ = 0.257), and global writing accuracy (F(2, 85) = 32.09, *p* < 0.001, η^2^_p_ = 0.430). Additionally, the analysis clearly showed significant main effects of severity for all measures (F(2, 88) = 22.76, *p* < 0.001, η^2^_p_ = 0.341 for global reading speed; F(2, 88) = 98.80, *p* < 0.001, η^2^_p_ = 0.692 for global reading accuracy; F(2, 85) = 73.86, *p* < 0.001, η^2^_p_ = 0.635 for global writing accuracy).

Tukey post-hoc analysis conducted on the severity groups revealed significant differences in performance in the target ability. Considering global reading speed, the analysis showed differences between Group a (less severely impaired, *n* =) and Group b (moderately impaired) (*p* < 0.001), Group a and Group c (severely impaired) (*p* < 0.001), and Group b and Group c (*p* < 0.001). Figure 1 shows performances in reading speed impairment severity groups, controlling for age.

For global reading accuracy, a difference was found between Group a (less severely impaired) and Group b (moderately impaired) (*p* < 0.001), Group a (less severely impaired) and Group c (severely impaired) (*p* < 0.001), and Group b (moderately impaired) and Group c (severely impaired) (*p* < 0.001). See Figure 2 for performances of reading accuracy impairment severity groups (with age as a covariate).

The same differences between impairment severity groups were found for global writing accuracy (*p* = 0.001, *p* < 0.001, *p* < 0.001, respectively for Group a and Group c, Group a and Group b, and Group b and Group c). Figure 3 shows performances in the different writing-accuracy impairment severity groups (age as covariate).

Setting age as a covariate in the same repeated measures ANOVAs did not change the results (treatment x severity: all *p* < 0.001).

In order to avoid spurious effects due to the use of Z-scores, which could result in excessively emphasizing smaller changes in speed or accuracy for older children (for whom the norms show greatly reduced standard deviations with respect to younger children) and to regression to the mean, all comparisons between severity groups yielding significant differences were further checked applying Oldham’s (1962) method, see [36]. The results showed that all improvements between the two time points significantly and positively correlated with the average of pre- and post-test measures (all r_s_ > 0.430, all *p* < 0.001). The differences in changes in reading speed/accuracy and in writing accuracy related to starting levels could thus not be due to regression to the mean. Table 2 shows descriptive statistics of the three impairment severity groups.

#### 3.1.3. Discussion

In the first part of the present study, we compared changes between pre- and post-test in older versus younger children and in more versus less severely impaired children in reading speed/accuracy and writing accuracy. Difference scores between the post-test and the pre-test, expressing training-related changes, were compared between groups of children. Considering the three age groups, no difference in improvement was observed for reading accuracy and speed, while a significant difference emerged for writing accuracy. Precisely, the group of younger children, below 9 years, improved more than the older children in writing accuracy.

The comparison among severity groups showed that, for reading speed, reading accuracy, and writing accuracy, the group of children with more severe initial impairments improved more than groups of children with less severe impairments, and that this was neither due to use of Z-scores producing spurious effects, nor to regression to the mean. A further step was to focus on follow-up performance, in order to evaluate the consolidation of pre-test to post-test gains.

### 3.2. Consolidation of the Improvements: Comparison of the Pre-Treatment Scores with Follow-Up Scores

#### 3.2.1. Comparison between Age Groups

In ANOVAs comparing pre-treatment and follow-up scores (treatment/consolidation as within factor), and age group as between factor, significant differences were found in global reading speed, global reading accuracy, and global writing accuracy (main effect of consolidation, reflecting the stability of obtained improvements), F(1, 77) = 64.22., *p* < 0.001, η^2^_p_ = 0.455; F(1, 77) = 13.85, *p* < 0.001, η^2^_p_ = 0.152; F(1, 76) = 24.40, *p* < 0.001, η^2^_p_ = 0.243, respectively). Post-hoc analysis showed significant consolidation effects for all measures in Group 1 (younger than 9 years) (all *p* < 0.014) and in Group 2 (between 9 and 10 years) (all *p* < 0.027). In Group 3 (11 years and older), the stability of improvements was confirmed for global reading speed (*p* < 0.001) and syllables/second in text reading (*p* < 0.001), while global reading accuracy and global writing accuracy were not significant (*p* > 0.23).

For global reading speed and global reading accuracy the analyses showed no significant consolidation x age interactions (all *p* > 0.41). However, a significant consolidation x age interaction emerged for global writing accuracy (*p* = 0.006). No significant age effects emerged (all *p* > 0.12).

Tukey post-hoc analysis was conducted on global writing accuracy, yet showed no significant differences between the three groups (all *p* > 0.13). Additionally, *t*-tests comparing pre-test to follow-up difference scores in the three groups failed to show any significant difference between pairs of groups (all *p* > 0.09).

#### 3.2.2. Comparison between Severity Groups

The consolidation of improvements from pre-test to follow-up was compared in the three severity groups. A significant consolidation effect emerged for global reading speed (F(1, 77) = 79.96, *p* < 0.001, η^2^_p_ = 0.509), global reading accuracy (F(1, 77) = 15.33, *p* < 0.001, η^2^_p_ = 0.166), and global writing accuracy (F(1, 76) = 43.87, *p* < 0.001, η^2^_p_ = 0.366). Consolidation x severity effects also emerged on the three measures (all *p* < 0.019). Moreover, as expected, a significant severity effect was found for all parameters (all *p* < 0.001).

Tukey post-hoc analysis conducted on the severity groups showed that improvements/consolidation on both global reading speed and global reading accuracy differed between Group a (less severely impaired) and Group b (moderately impaired) (*p* < 0.001), Group a and Group c (severely impaired) (*p* < 0.001), and Group b and Group c (*p* < 0.001). Lastly, global writing accuracy appeared different between Group a (less severely impaired) and Group b (moderately impaired) (*p* = 0.001), Group a and Group c (severely impaired) (*p* < 0.001), and Group b and Group c (*p* < 0.001).

Setting age as a covariate in the same repeated-measures ANOVAs did not change these results (treatment x severity, all *p* < 0.001).

Additional analyses following Oldham’s method showed significant correlations between the average of pre-test and follow-up measures and the improvements in global reading speed and global writing accuracy (all r_s_ > −0.335, all *p* < 0.003), while global reading accuracy did not reach significance (r = −0.145, *p* = 0.201). In this case, improvements in performances in the three impairment severity groups based on reading accuracy performance could be partially explained by a spurious effect due to regression to the mean.

#### 3.2.3. Discussion

In this second part of the present study, we evaluated the consolidation of the improvements obtained at six months after the end of Tachidino treatment in different groups of children. We observed the stabilization of the improvements obtained in the post-test assessment. Specifically, upon comparison of the three age groups, it was observed that children of the youngest group obtained a greater improvement in writing accuracy with respect to older children.

Considering the differences between impairment severity groups, results showed that, for reading speed, reading accuracy, and writing accuracy, the group of children with more severe initial impairments at pre-test improved more than groups of children with less severe impairments.

### 3.3. Treatment Related Changes: Effects of Discontinuation of Treatment between Post-Test and Follow-Up

#### 3.3.1. Comparison between Age Groups

Comparing post-test and follow-up results, a significant difference on global reading accuracy (F(1, 77) = 18.19, *p* < 0.001, η^2^_p_ = 0.191) emerged, while the other two global measures did not differ between the two time points (*p* > 0.23). In particular, a post-hoc analysis showed differences in global reading accuracy within the three groups (all *p* < 0.008). Neither significant discontinuation x age interactions were found (all *p* > 0.32), nor any age effects (all *p* > 0.42). Figure 4 shows performances at pre-test, post-test, and follow-up of the three age groups.

#### 3.3.2. Comparison between Severity Groups

Regarding possible changes due to discontinuation of treatment between post-test and follow-up, the two performances were compared across impairment severity groups. Global reading speed and global writing accuracy did not differ between post-test and follow-up assessments (all *p* > 0.28), while a significant difference was found for global reading accuracy (F(1, 77) = 21.68, *p* < 0.001, η^2^_p_ = 0.220).

Discontinuation x severity interaction effects were not significant (all *p* > 0.06).

Specific severity effects, as it could be predicted, were found for global reading speed (F(2, 77) = 68.15, *p* < 0.001, η^2^_p_ = 0.639), global reading accuracy (F(2, 77) = 49.63, *p* < 0.001, η^2^_p_ = 0.563), and global writing accuracy (F(2, 76) = 37.36, *p* < 0.001, η^2^_p_ = 0.496). Figure 5 shows performances at pre-test, post-test, and follow-up of the three severity groups.

Additionally, post-hoc analyses showed that the differences in performance level among groups remained significant after treatment (for all pairs, *p* < 0.005). Setting age as a covariate in the same repeated measures ANOVAs did not change the results (treatment x severity all *p* > 0.056).

#### 3.3.3. Discussion

In the last analyses on possible effects of the discontinuation of treatment on improvements, comparing post-test and follow-up scores, no interaction between discontinuation and severity was found. Global reading accuracy was the only measure that differed between the two time points, both for age groups and impairment severity groups comparisons. We hypothesized that one reason why reading accuracy shows a certain decrease in improvement after post-test is that the reading accuracy score includes text reading, for which different passages are provided for each grade, thus posing greater difficulties to children who had passed to the following grade at follow-up test. In other terms, since the typology and difficulty level of the text reading test increased from one grade to the next grade, children received a different, more complex text at follow-up with respect to pre- and post-test assessments.

To address this hypothesis, a follow-up analysis was performed decomposing the global accuracy score into its component scores. Paired t-tests on Z-scores confirmed that the worsening from post-test to follow-up did not concern word reading accuracy (t (79) = 0.798, *p* = 0.427), while it did concern text reading accuracy (t (78) = 5.475, *p* < 0.001). Nonetheless, decrease in accuracy after post-test also concerned nonword reading (t (78) = 3.624, *p* = 0.001), thus our hypothesis was only partially confirmed. This suggests that reading accuracy is indeed a parameter that suffers more from discontinuation in intervention, and/or longer treatment periods should be foreseen in order to obtain more stable improvements in this respect.

## 4. General Discussion and Conclusions

The aim of the present study was to compare the beneficial gains in reading and in writing among different groups of children with a diagnosis of developmental dyslexia. Specifically, we assessed possible age-related differences in the outcomes of the treatment, and the impact of the severity of the impairments on the gains. To this aim, participants were subdivided into three different age groups representing different stages of reading acquisition, and three groups characterized by different levels of impairment with respect to reading speed, reading accuracy, and writing accuracy, in turn. All children received a one-month treatment with Tachidino, a web-based program for neuropsychological intervention in developmental dyslexia [7].

In the first part of the study, treatment-related changes were assessed. Considering the three age groups, no difference in improvement was observed for reading accuracy and speed, while a significant difference emerged for writing accuracy. Specifically, the group of younger children, below 9 years of age, improved more than older children in this measure. This was not due to reduction of effectiveness at older age (improvement of writing skills as measured in z-scores were high in all age groups) but rather to a particularly high rate of improvement in the youngest group.

The comparison among impairment severity groups showed that, for all measures, i.e., reading speed, reading accuracy, and writing accuracy, children with more severe initial impairments improved more than groups of children with less severe impairments. This was additionally confirmed after checking for possible spurious effects due to regression to the mean and, for reading speed only (for which raw scores had been recorded as syllables per second), to the use of Z-scores possibly amplifying gains in older children.

In recent years, and mostly during the last months of the COVID-19 pandemic, the use of remote interventions has made it possible to keep rehabilitation programs active, reducing the costs of treatments while increasing the efficiency of reading remediation programs [22,37]. Tachidino is a multi-componential treatment which incorporates two different stimulations: Visual-Attentional Training [8,9] and Visual Hemisphere-Specific Stimulation (VHSS) [10,11]. The effectiveness of the two component trainings had already been shown [9,12,13,16,17,18] as had the effectiveness of an intensive combined treatment [38]. The present study, using a remote multi-componential treatment where both components are not only added to each other but merged into a single treatment, confirms the enhancements described in Cancer and colleagues [38], which adopted an outpatient remediation program.

In the second part of the study, we explored the consolidation of reading and writing improvements six months after the discontinuation of treatment. Moreover, in that case, different groups of children were compared. The results substantially confirmed those obtained in the post-test assessment. Regarding the comparison of the three age groups of children, the younger children obtained, and maintained, a greater improvement in writing accuracy with respect to both groups of older children. The comparison between the severity groups showed that for all measures, i.e., reading speed, reading accuracy, and writing accuracy, the group of children with more severe initial impairments obtained larger gains than groups of children with less severe impairments. Possible reduction in improvements between post-test and follow-up were also investigated in the different groups. Reading accuracy was the only measure for which it was possible to observe a worsening of performance from post-test to follow-up assessment. It should be highlighted, though, that the difference between pre-test and follow-up was still highly significant, indicating that improvement was maintained and still relevant, even if its size was reduced from post-test to follow-up. These results are in line with those of previous studies showing that remote treatment of dyslexia may improve reading speed and accuracy after only a few weeks of treatment, fostering automatization of the reading process [22]. In addition, a recent study on the effects of computerized cognitive training on visual-spatial working memory and reading performance showed that improvements in attention and visual-spatial working memory lasted for a period of 6 months after treatment [39].

The main findings of the present work relate to the absence of any evidence of reduction in treatment effectiveness for children who are either older or severely impaired. By contrast, older children were found to improve more than younger children in reading speed when using raw scores comparisons. This confirms that there are no objective reasons for limiting treatment to younger children, as what often occurs in clinical practice, nor to children with less severe impairments. The idea that younger children are more responsive to treatment is probably derived from older theories about the reduction in brain plasticity after a certain age, and from research on language impairments, where indeed there is evidence of a developmental window (the so-called “critical period”) when response to intervention is more likely to occur and to a larger extent [40]. However, it should be considered that language is a largely different function compared to reading and writing abilities. Indeed, language development follows stable, pre-determined trajectories that are linked to neurobiological development, although it additionally reflects the effects of social interaction and, minimally, of direct instruction [41]. Reading and writing, by contrast, are lately (both phylogenetically and ontogenetically) learned abilities substantially related to formal education, therefore their development is not biologically determined and, consequently, can reasonably be more flexible and subjected to adaptation to environmental conditions. On these grounds, it is not surprising that similar improvements can take place even at later stages of development, such as adolescence, as was also observed in adult populations [26,30,31].

Regarding severity, clearly it could be argued that the more impaired an ability is at an initial stage, the more space there is for improvement. This line of reasoning is probably the simplest explanation for our findings about greater improvement in more severely impaired children. Nonetheless, it should be noted that, in spite of this statistical truth, more severely impaired children are often not offered intervention in clinical practice, favoring other forms of support for their education and academic development (e.g., technological aids to support reading and writing etc.). This is of course justified if observing that a very severe impairment makes it difficult to access school contents through written sources and to express oneself in written form. In this perspective, it is perfectly reasonable that these children should be offered more support and more aids in their school education. This reasoning, though, does not imply that intervention targeted to improve their reading and writing skills, in addition to external and technological support, should be useless or irrelevant. Looking at the large improvements that were observed in these children’s scores after only one month of treatment, it is evident that such changes are highly relevant in both absolute and relative terms (as shown by raw scores and z-scores), and that these children are much closer to “normal” performance at post-test than they were at pre-test. Even a change from a very severe impairment to a moderate or non-severe impairment could allow these children to access tasks and activities or experiences that were precluded before treatment, and affect their self-perceptions and self-confidence to a large extent, improving their general quality of life [42,43]. Moreover, further research should clarify whether repeated cycles of treatment could bring further, clinically relevant improvement: existing studies indicate that improvement after the first cycle of intervention is reduced as compared to the former, but still significant. What the present results allow us to exclude quite categorically is that the presence of a more severe impairment reduces the effectiveness of treatment, at least if treatment is conducted according to multi-factorial principles and involves multiple functions. It is possible that other studies which suggested a reduced response to intervention in severely impaired children employed different types of interventions that acted on mechanisms which become less responsive and more rigid over time, or that acted on multiple components allowing for compensatory processes to take place based on different, distributed networks of functions [5,6].

Clearly, the present study does not provide answers to all the questions that were raised, and more research is needed to clarify, for instance, the effects of longer treatments for very severe disorders, or the persistence of improvements after longer periods of discontinuation of treatment. Moreover, some theoretical questions remain open. First of all, the effectiveness observed from the Tachidino program may depend on both components of the training and it is not possible to disentangle the two contributions. We assumed that each of the two components of the treatment produced effects that are similar to those observed in previous studies when each component was applied separately, i.e., relatively larger effects on accuracy through the VHSS [12,13,15,17] and larger effects on speed through the AVG [9,16,18,20] component. The precise, specific contribution of auditory stimulation would need to be investigated through controlled experiments. A comparison of the results of the intervention program with other treatment programs is presented elsewhere (Lorusso et al., submitted). The absence of a direct comparison of the improvements with those obtained in a control group in the present study constitutes a limitation, nonetheless the stability of improvements at follow-up and in spite of reading different texts (as shown by the comparison of raw scores in text reading tests) suggests that improvement itself can hardly be the result of simple test repetition effects.

Altogether, we believe that the data presented provide a very clear-cut picture of differential, age-related and severity-related constraints on reading and writing improvements, and that these should be considered in clinical practice planning and investments. Even if the present results would require replication and the effectiveness of the Tachidino program has to be demonstrated in comparison with other interventions, the present evidence is highly encouraging with respect to the possibility of a large-scale, even remotely delivered intervention for these highly frequent disorders (3–5% in the Italian school population [44].

## Figures and Tables

**Figure 1 children-09-00071-f001:**
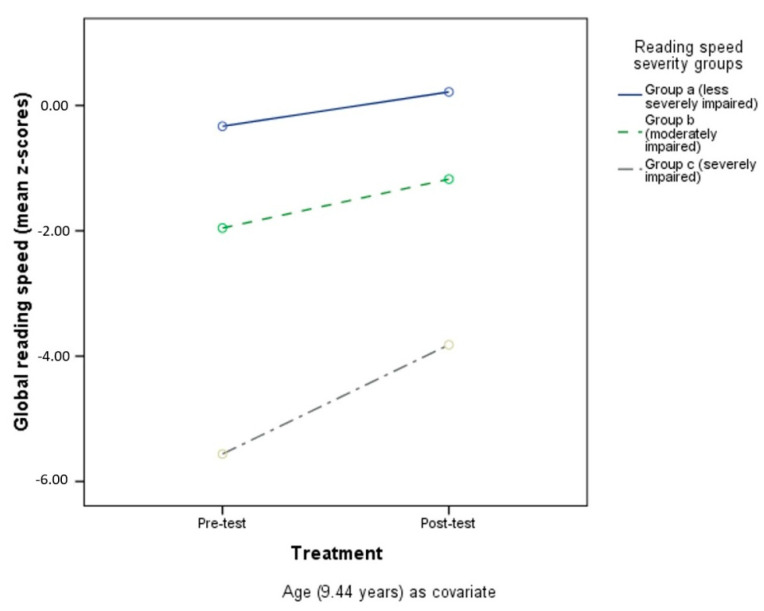
Pre- and post-test reading speed in the three different impairment severity groups.

**Figure 2 children-09-00071-f002:**
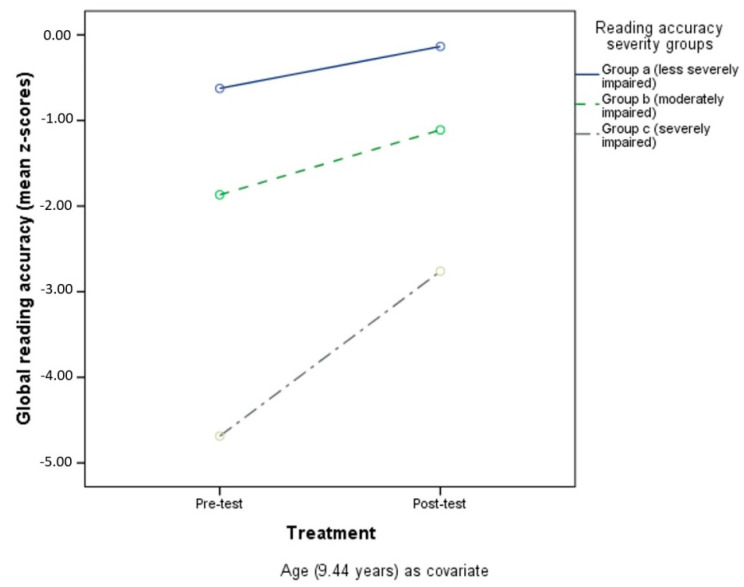
Pre- and post-test reading accuracy in the three different impairment severity groups.

**Figure 3 children-09-00071-f003:**
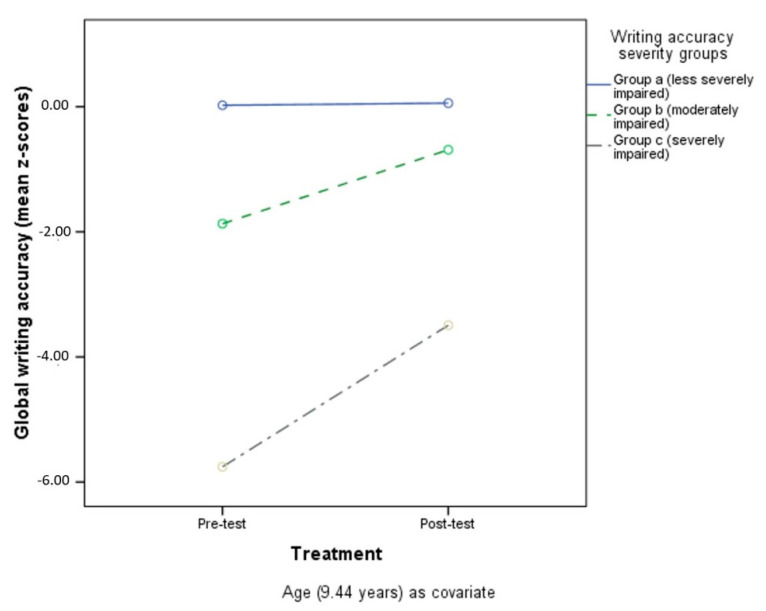
Pre- and post-test writing accuracy in the three different impairment severity groups.

**Figure 4 children-09-00071-f004:**
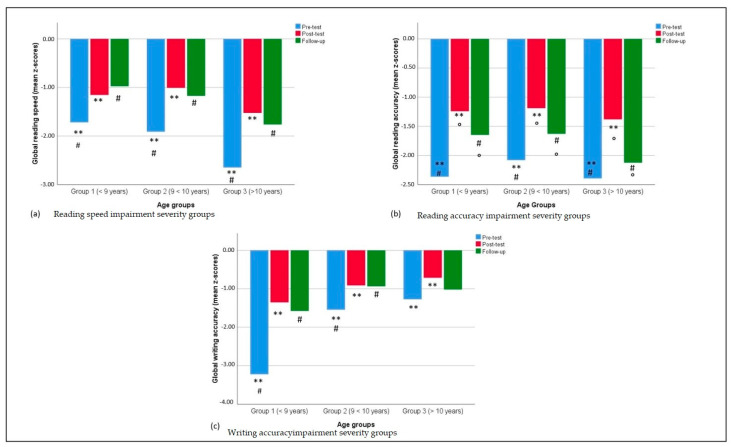
Pre-test, post-test, and follow-up reading and writing scores in the three different age groups. Identical symbols (**, °, #) indicate pairs of significantly different variables within each between-groups comparison.

**Figure 5 children-09-00071-f005:**
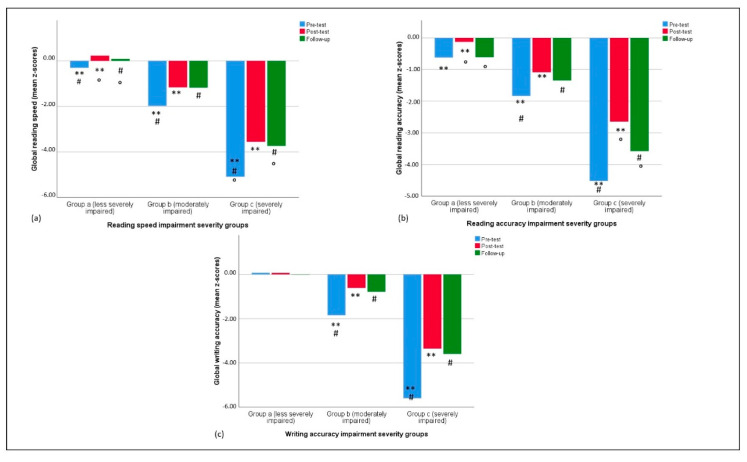
Pre-test, post-test, and follow-up reading and writing scores in the three different impairment severity groups. Identical symbols (**, °, #) indicate pairs of significantly different variables within each between-groups comparison.

**Table 1 children-09-00071-t001:** Mean scores (SD in parentheses) and effect sizes (Cohen’s d_z_) of the three age groups.

		Group 1 (Younger than 9 Years)		Group 2 (between 9 and 10 Years)		Group 3 (11 Years Old and Older)	
		Mean (SD)	Cohen’s d_z_ (*p*)	Mean (SD)	Cohen’s d_z_ (*p*)	Mean (SD)	Cohen’s d_z_ (*p*)
Global reading speed	PRE	−2.40 (2.96)	−0.77 (<0.001)	−1.95 (2.01)	−0.99 (<0.001)	−2.61 (2.34)	−0.90 (<0.001)
	POST	−1.61 (2.09)		−1.05 (1.79)		−1.51 (1.75)	
Global reading accuracy	PRE	−2.81 (1.97)	−1.19 (<0.001)	−2.05 (1.80)	−0.80 (<0.001)	−2.47 (1.92)	−0.71 (0.003)
	POST	−1.47 (1.28)		−1.17 (1.32)		−1.51 (1.73)	
Global writing accuracy	PRE	−3.68 (2.91)	−1.46 (<0.001)	−1.53 (2.84)	−0.40 (0.015)	−1.22 (1.53)	−0.56 (0.019)
	POST	−1.73 (2.38)		−0.92 (2.24)		−0.70 (1.45)	

**Table 2 children-09-00071-t002:** Mean Z-scores (SD in parentheses) and effect sizes (Cohen’s d_z_) of the three different impairment severity groups (NB for each comparison, the severity of impairment is established based on the parameter being compared: reading speed, reading accuracy, writing accuracy, in turn).

		Group a (Less Severely Impaired)		Group b (Moderately Impaired)		Group c (Severely Impaired)	
		Mean (SD)	Cohen’s d_z_ (*p*)	Mean (SD)	Cohen’s d_z_ (*p*)	Mean (SD)	Cohen’s d_z_ (*p*)
Global reading speed	PRE	−0.33 (0.46)	−0.88 (<0.001) (*n* = 26)	−1.96 (0.57)	−0.86 (<0.001) (*n* = 46)	−5.56 (3.24)	−1.38 (<0.001) (*n* = 19)
	POST	0.20 (0.63)		−1.17 (0.99)		−3.81 (2.14)	
Global reading accuracy	PRE	−0.63 (0.31)	−1.06 (<0.001) (*n* = 24)	−1.87 (0.48)	−1.06 (<0.001) (*n* = 40)	−4.69 (1.77)	−1.20 (<0.001) (*n* = 27)
	POST	−13 (0.48)		−1.12 (0.75)		−2.75 (1.53)	
Global writing accuracy	PRE	0.05 (0.54)	0.04 (0.83) (*n* = 37)	−1.87 (0.63)	−1.30 (<0.001) (*n* = 28)	−5.79 (2.76)	−1.42 (<0.001) (*n* = 23)
	POST	0.02 (0.86)		−0.68 (1.01)		−3.44 (2.76)	

## Data Availability

Due to privacy issues and as requested by the Ethics Committee, data will be made accessible upon request to the first author and under appropriate agreement conditions.

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
