# Peer review of "Remote Neuropsychological Intervention for Developmental Dyslexia with the Tachidino Platform: No Reduction in Effectiveness for Older Nor for More Severely Impaired Children"

_children, 2022, doi:10.3390/children9010071_

Round 1

Reviewer 1 Report

The paper examines the effect of a rehabilitation program (Tachidino) on the reading and writing deficits of a moderately large group of children with dyslexia (of varying age and severity).  The study is generally well-conducted, and the results are interesting.  I feel that the paper can be of interest to all researchers and clinicians dealing with learning disorders.  Below, I detail a few aspects of the manuscript which I feel may be improved in a revision of the paper.

The authors chose to use a rehabilitation program that mixes up two different types of training, one based on Visual-Attentional Training and one based on the Visual Hemisphere-Specific Stimulation (VHSS) developed according to Bakker’s Balance Model.

I have a few comments on this choice and its implications.

First, I find the description on page 4, lines 166-184, a bit difficult to follow.  I gather that the logic is that the two trainings are mixed, in the sense that in each trial the child has to perform a visual-spatial task as well as a word recognition task.  Is that correct?  If so, maybe the authors could try to make this point more clearly. 

Second, some further aspects of the methodology do not seem entirely clear to me. 

  1. The authors state that children spent in training an average total time of 14 hours over a maximum of four weeks. However, I do not find indications of the amount of time per session and, more importantly, of the number of trials per session. This information could be important to understand the actual amount of exercise carried out by children.
  2. As I understand it, in each trial the children fixated the center of the screen and the word appeared on either side of the screen. The VHSS is based on the original idea by Bakker that different types of subjects (L- or P-types) should receive different hemispheric stimulations. Was this carried out in the present study?  The authors do not mention it. More generally, they do not say whether presenting words to the left or right fields did make a difference.  This seems important to try to understand whether the amelioration effect was due to A) the matching between L- or P-type and type of stimulation, B) the side of the stimulation (eg right hemifield more than left), or C) simply to the use of tachistoscopic stimulation.  I feel that some clarifications should be given on this point.  Using successfully the VHSS may superficially appear to confirm the model by Bakker.  However, further information is needed to understand whether this was the case or simply the methodology used was loosely derived from the original Bakker’s proposal.
  3. I am not entirely clear what is the rationale for using an auditory presentation in a part (half?) of the trials. Since the children had a reading deficit, it is not immediately apparent what could be the idea underlying this part of the procedure.  I would ask the authors to clarify this aspect.

Third, whatever is the rationale underlying Bakker’s part of the program, mixing it with the Visual-Attentional Training might reasonably have had the effect of improving the efficacy of the training.  This seems a good choice from a clinical and rehabilitative perspective.  On logical grounds, one may wonder to what extent the observed improvements were due to the different parts of the training.  The authors mention this point in the discussion but maybe it should be dealt with more explicitly.

Another aspect that I think should be considered in the Discussion section is the type of rehabilitation design used in the study.  Apart from the follow-up part of the research, the authors simply looked at pre-post differences.  There was no assessment of the stability of the reading deficit before the start of the study as well as no control group with no treatment or alternative types of treatment.  This choice may be understandable on clinical grounds and it does not necessarily undermine the significance of the research.  However, some comments on the limitations of the research connected to the study design may be in order.

Reviewer 2 Report

The paper is definitely interesting, well written and well referenced. The theoretical background is solid as well as the research design and the discussion effectively convey the significance of the results. The intervention described might be a viable and feasible solution for clinical practice and it is worth spreading (and particularly relevant for the special issue dedicated to Technology in Rehabilitative Interventions). The investigation of two important moderators of the treatment effect (age and severity) is noteworthy and represent a major strength of the research.

A couple of misprints should probably be revised. For example,  "reading speed severity", "reading accuracy severity", and "writing accuracy severity" could be reported as "reading/writing speed/accuracy problem severity" or similar (lines 246-248).  Then, the authors wrote "A significant Treatment x Age effect was found for [...]", but I guess they meant "Treatment x Severity effect" (lines 251-252) since the paragraph is regarding the treatment efficacy for severity groups.

With respect to the research methodology, I would like to share a few points with the authors. A power analysis was conducted to determine desired sample size (althought it is reported as sensitivity analysis, I guess the purpose of the analysis was to determine the required sample size to obtain a given effect size and not to obtain a required effect size) and the authors set a value of 0.17 for the effect size to detect. It is not clear to me whether this value is derived from other researches in the field or they considered conventional guidelines as those proposed by Cohen (1988) as reference.

Dealing with effect sizes, I appreciated that the authors reported an effect size measure that control for intra-subjects variability like partial eta-squared. Even so, I was wondering whether they considered reporting also an effect size measure derived from standardized mean differences and feasible for within-subjects design like Cohen's d-z (see for example the very useful primer by Lakens, 2013) at least for pre-post mean difference on the whole group of subjects. Otherwise, pre- and post- means and standard deviations could be reported in order to make it easy to calculate such an effect size and to simplify the inclusion of the results of the study in future meta-analyses.

Anyway, the authors did a very good job with this paper.
